# Complete Revascularization and One-Year Survival with Good Neurological Outcome in Patients Resuscitated from an Out-of-Hospital Cardiac Arrest

**DOI:** 10.3390/jcm11175071

**Published:** 2022-08-29

**Authors:** Vilma Kajana, Enrico Baldi, Francesca Romana Gentile, Sara Compagnoni, Federico Quilico, Luca Vicini Scajola, Alessandra Repetto, Alessandro Mandurino-Mirizzi, Marco Ferlini, Barbara Marinoni, Maurizio Ferrario Ormezzano, Roberto Primi, Sara Bendotti, Alessia Currao, Simone Savastano

**Affiliations:** 1Division of Cardiology, Fondazione IRCCS Policlinico San Matteo, Viale Golgi 19, 27100 Pavia, Italy; 2Humanitas Mater Domini, 21053 Castellanza, Italy; 3Department of Molecular Medicine, University of Pavia, 27100 Pavia, Italy

**Keywords:** out of hospital cardiac arrest, survival, multivessel disease, complete revascularization, percutaneous coronary intervention

## Abstract

**Background.** The survival benefit of complete versus infarct-related artery (IRA)-only revascularization during the index hospitalization in patients resuscitated from an out-of-hospital cardiac arrest (OHCA) with multivessel disease is unknown. **Methods.** We considered all the OHCA patients prospectively enrolled in the Lombardia Cardiac Arrest Registry (Lombardia CARe) from 1 January 2015 to 1 May 2021 who underwent coronary angiography (CAG) at the Fondazione IRCCS Policlinico San Matteo (Pavia). Patients’ prehospital, angiographical and survival data were reviewed. **Results.** Out of 239 patients, 119 had a multivessel coronary disease: 69% received IRA-only revascularization, and 31% received a complete revascularization: 8 during the first procedure and 29 in a staged-procedure after a median time of 5 days [IQR 2.5–10.3]. The complete revascularization group showed significantly higher one-year survival with good neurological outcome than the IRA-only group (83.3% vs. 30.4%, *p* < 0.001). After correcting for cardiac arrest duration, shockable presenting rhythm, peak of Troponin-I, creatinine on admission and the need for circulatory support, complete revascularization was independently associated with the probability of death and poor neurological outcome [HR 0.3 (95%CI 0.1–0.8), *p* = 0.02]. **Conclusions.** This observation study shows that complete myocardial revascularization during the index hospitalization improves one-year survival with good neurological outcome in patients resuscitated from an OHCA with multivessel coronary disease.

## 1. Introduction

Out-of-hospital cardiac arrest (OHCA) is a leading cause of mortality worldwide [1,2]. The annual incidence of OHCA in Europe is between 67–170 per 100.000 inhabitants with a survival rate of about 8% at discharge [1,2,3,4]. Eighty percent of OHCA have a cardiac etiology, of which seventy percent are a consequence of acute coronary syndromes (ACS) [5]. The electrocardiogram (ECG) acquired after the return of spontaneous circulation (ROSC) represents the cornerstone for the identification of an ST-elevation suggestive of acute myocardial infarction [6,7]. 

Survival after OHCA represents the most important challenge both in the whole OHCA population and in the subset of OHCA patients in which the ACS represents the underlying cause of cardiac arrest. Literature is agreed in saying that in patients with OHCA due to an acute myocardial infarction, early revascularization is associated with a better outcome [8,9]. Jentzer et al. showed that early percutaneous coronary intervention (PCI) was associated with better survival in patients with OHCA both in the setting of ST-elevation myocardial infarction (STEMI) and non-ST-segment elevation myocardial infarction (NSTEMI) [8]. Similarly, Tanberg et al. highlighted that an early PCI was associated with a higher survival rate in patients with OHCA of supposed cardiac aetiology [9]. Because of the known benefit of revascularization in this setting, the European Society of Cardiology (ESC) guidelines recommend urgent coronary angiography (CAG) in patients after OHCA in whom the electrocardiogram (ECG) demonstrates a STEMI or haemodynamic/electrical instability. Moreover, they also suggest to consider an urgent CAG in OHCA patients with high suspicion of ongoing ischaemia [10]. However, OHCA patients frequently have multivessel disease highlighted at CAG [11]. Data from different studies has shown that 35–45% of OHCA patients have a multivessel disease: 12–20% have a two-vessel disease and 15–30% have a three-vessel disease [11,12,13]. Considering patients with STEMI and a multivessel disease at CAG, it is still a matter of debate if a strategy of complete revascularization during the index hospitalization rather than a staged multivessel PCI should be preferred. The same is true in those STEMI patients presenting with cardiogenic shock. The complete revascularization during the index hospitalization is indeed only a class IIa recommendation and the best timing (whether during the index hospitalization or after patient’s discharge) of the management of non-infarct related artery (IRA) lesions remains a gap of knowledge, highlighted by the guidelines [10]. Similarly, in patients presenting with OHCA there is not a clear recommendation on the preferred type of revascularization strategy (complete versus IRA-only revascularization during the index hospitalization) [10] and there are no data regarding long-term outcomes. We aimed to assess whether a complete revascularization during index hospitalization could lead to a better one-year survival with good neurological outcome in patients resuscitated from an OHCA and affected by multivessel disease.

## 2. Material and Methods

### 2.1. Type of Study

This is an observational study based on a retrospective analysis of prospectively collected data. 

### 2.2. Patient Selection

All the consecutive patients resuscitated from OHCA in the province of Pavia from 1 January 2015 to 1 May 2021 who were transported to the “Fondazione IRCCS Policlinico San Matteo”, the tertiary care hospital of the province of Pavia, and underwent a CAG during the index hospitalization following the OHCA were enrolled in the study. The decision to perform a CAG was left to the admitting physician and the decision to perform a complete revascularization or an IRA-only revascularization during the index procedure was left to the interventional cardiologist. After the index procedure the decision to provide the patient with a complete revascularization was taken jointly together with clinical cardiologists and cardiac surgeons in case of three vessels or left main disease.

### 2.3. Definitions

OHCA was defined as the cessation of cardiac mechanical activity, confirmed by the absence of signs of circulation that occurred outside of a hospital setting.

The ECGs were categorized as diagnostic for STEMI (or not) according to the criteria for the electrocardiographic diagnosis of STEMI recommended by the 2017 guidelines of the European Society of Cardiology (ESC) [10].

A coronary artery stenosis was defined as significant if it was greater than 50% for the left main coronary artery (LMCA) and greater than or equal to 70% for the other coronary vessels [14,15].

Complete revascularization (CR) was considered the percutaneous or surgical treatment of all the significant coronary artery stenosis before hospital discharge, whilst IRA-only was considered the treatment of only the culprit artery during the index hospitalization. 

The neurologic outcome was evaluated according to the Cerebral Performance Category (CPC) scale: CPC values of 1 or 2 were considered as good neurological outcome, whilst CPC values more than 2 as bad neurological outcome

Survival at discharged was defined at the moment of hospital discharge or at 30 days of hospitalization as indicated by the Utstein style on which our registry in based.

### 2.4. Study Endpoints

One-year survival with good neurological outcome was the primary endpoint. The secondary endpoint was one-year survival with good neurological outcome with consideration only to patients discharged alive.

### 2.5. Data Management 

After being anonymized, study data were collected and managed using REDCap (Research Electronic Data Capture, ver 12.0.27 created by Vanderbilt University, Nashville, TN, USA) hosted at the Fondazione IRCCS Policlinico San Matteo [16,17]. REDCap is a secure, web-based application designed to support data capture for research studies, providing (1) an intuitive interface for validated data entry; (2) audit trails for tracking data manipulation and export procedures; (3) automated export procedures for seamless data downloads to common statistical packages; and (4) procedures for importing data from external sources.

### 2.6. Data Collection

Patients’ and OHCA characteristics were collected according to the 2014 Utstein style recommendations [18]. The pre-hospital data related to the OHCA were obtained from the Cardiac Arrest Registry of the Lombardy Region, named Lombardia CARe, which was approved by the ethical committee of the Fondazione IRCCS Policlinico San Matteo in Pavia and registered on ClinicalTrial.gov (NCT03197142). To be enrolled in the Lombardia CARe the signature of an ad-hoc informed consent is required. The signature is acquired during hospital stay or during follow-up according to the neurological recovery of the patients. In case of comatose patients the admitting physician decides about the enrollment and a signature of a legal representative is successively required. The informed consent to the interventional procedures is required separately and it is signed by the patients only if he/she is awake and out of an emergency condition.

Survival data and time to death were also collected. Patients’ mortality status as well as neurological outcome were analyzed at discharge or at a 30-day follow up. Clinical data were retrieved from the medical report of the index hospitalization.

### 2.7. Statistical Analysis

We calculated the required sample size for the comparison of survival rates in two independent groups. Considering an estimated one-year survival with good neurological outcome after hospital discharge of 90% [19] and a 30% higher survival in the complete revascularization group in the general population [20] and assuming a type 1 error of 5% and a type 2 error of 20%, the required sample was 74 patients in case of a 1:1 ratio and 102 patients in case a 1:2 ratio (34/68).

Categorical variables were compared with the Chi-square test and presented as number and percentage. Continuous variables were compared with the *t*-test and presented as mean ± standard deviation or compared with the Mann-Whitney test and presented as median and interquartile range (IQR) according to normal distribution tested with the D’Agostino-Pearson test. Uni- or multivariable logistic regression were run to assess the association between one binomial dependent variable and one or more not correlated independent variables. Uni- and multivariable Cox regression models were performed to analyze the effect of one or more independent factors on the hazard of the combination of death or unfavourable neurological outcome. Only statistically significant and noncorrelated variables in univariable analysis were inserted in the multivariable model. Survival curves were created according to the Kaplan–Meier method and compared with the log-rank test. Statistical analyses were performed with MedCalc (version 20.106 64 bit, MedCalc Software Ltd., Ostend, Belgium). A *p* value < 0.05 was considered statistically significant.

## 3. Results

### 3.1. Patient Characteristics

Throughout the study period, 4982 OHCAs occurred in the province of Pavia and were therefore prospectively enrolled in the Lombardia CARe: 554 were successfully resuscitated and 400 were admitted at the Fondazione IRCCS Policlinico San Matteo hospital in Pavia. We then excluded 161 patients who did not undergo a coronary angiography either due to traumatic or clearly non-ischemic cause of arrest getting to the final amount of 239 patients who underwent a coronary angiography of which 119 showed a multivessel disease (Table 1).

The characteristics of resuscitated OHCA patients with multivessel disease are presented in Appendix A, whilst the characteristics of resuscitated OHCA patients with multivessel disease discharged alive are presented in Appendix A.

### 3.2. Complete Revascularization vs. IRA-Only Revascularization

Among the patients with multivessel coronary disease, complete revascularization was performed in 37 patients (31.1%), of whom 8 underwent the procedure during the index procedure and 29 did so in a staged procedure after a median time of 5 days [IQR 2.5–10.3]. An IRA-only revascularization was performed in 82 (68.9%) patients (Figure 1). 

Table 2 compares the two groups (complete vs. IRA-only revascularization).

On univariable analysis, the probability of receiving a complete revascularization was positively associated with the presence of a shockable presenting rhythm at presentation (OR 5.10; CI 95% 1.12–22.9, *p* = 0.03), and inversely related with cardiac arrest duration (OR 0.4, CI 95% 0.2–0.9, *p* = 0.02), the presence of a chronic total occlusion (OR 0.37; CI 95% 0.16–0.85, *p* = 0.02) and the level of serum creatinine (OR 0.16; CI 95% 0.04–0.7, *p* = 0.018). At multivariate analysis only the presence of chronic total occlusion maintained a statistically significant inverse association with the complete revascularization strategy (OR 0.4; CI 95% 0.2–0.98 *p* = 0.05) (Table 3).

### 3.3. Primary and Secondary Endpoints: Complete Revascularization and One Year Survival with Good Neurological Outcome

One-year survival was significantly higher in the complete revascularization group as compared to the IRA-only group (86.5% vs. 51.2%, *p* = 0.0003). This difference was even more relevant when considering our primary endpoint: one-year survival with good neurological outcome (CPC ≤ 2) in the overall population (83.3% vs. 30.4%, *p* < 0.001). Importantly, Kaplan-Meier survival curves diverged significantly at one-year survival with good neurological outcome (Figure 2). 

Table 4 shows the results of both univariable and multivariable Cox regression analyses for the probability of death or poor neurological outcome. After correction for cardiac arrest duration, shockable rhythm at presentation, peak of serum Troponin I, serum creatinine on admission and the need for pharmacological or mechanic circulatory support, complete revascularization was found to be independently and inversely associated with the probability of death and poor neurological outcome [HR 0.33 (95%CI 0.1–0.9), *p* = 0.04].

When considering our secondary endpoint we found that survival with good neurological outcome was still higher in the complete revascularization group (100% vs. 83.2%, *p* = 0.013) also when considering patients alive at hospital discharge. Kaplan-Meier curves contnued diverging significantly even after discharge (Figure 2B).

## 4. Discussion

The main finding from our study is that complete revascularization during the index hospitalization in rescued OHCA patients with multivessel disease at CAG is associated with a better one-year survival with good neurological outcome considering both the overall population and the patients discharged alive. Moreover, complete revascularization was shown to be independently associated with a reduced risk of death or poor neurological outcome at one-year.

Survival after OHCA with a cardiac aetiology remains low and multivessel coronary artery disease has been associated with a poor outcome [4,21]. Myocardial revascularization of the IRA artery is associated with a better outcome not only in the context of STEMI, but in all the OHCAs with an ischemic origin [8,10,22]. In a study of 599 OHCA patients written by Jentzer et al. [8], early CAG with PCI was found to be associated with improved survival compared to early CAG without PCI (65.5% vs. 45.5%). The authors also confirmed that early myocardial revascularization was associated with more than double the odds of a good neurological outcome compared to those with CAG alone or those who did not undergo CAG at all [8]. Moreover, in the same study, early CAG and early PCI were not associated with an increased incidence of acute kidney injury and transfusions [8]. Since about one-third of all the OHCA patients had at least two-vessel disease as seen on CAG following OHCA [8,9,10,11], the optimal management of this subgroup of patients is crucial. Such a finding, which correlates to our population, leads to the question if complete revascularization prior to hospital discharge would be beneficial when compared to PCI of the IRA alone. Some large, randomized trials have investigated the benefit of complete revascularization compared to IRA-only revascularization during the index hospitalization, but they have focused only on patients suffering from STEMI [20,23,24]. Among these, the most recent COMPLETE Trial demonstrated that in patients with STEMI and multivessel coronary artery disease, complete revascularization was superior to IRA-only revascularization in reducing the risk of cardiovascular death or myocardial infarction (7.8% vs. 10.5%) [20]. However, most of the largest trials excluded patients with OHCA, and therefore there is no evidence if this benefit is also present in this specific population, which has a very different pathophysiological characteristics compared to STEMI patients without cardiac arrest. There are only two studies on this specific topic published this year. The first is a single centre retrospective study by Chen and colleagues focused on a very small population and providing a very short follow-up limited to hospital discharge [25]. They showed that in-hospital mortality among cardiac arrest survivors with multivessel lesions is lower if complete revascularization is performed. The second is a multicentre retrospective study by Kim et al [26] reporting data from a population similar in size to our, although multicentre, with a follow up but still limited to one month after hospital discharge. They also found that a complete revascularization strategy allows better survival. As compared to these two studies we provide a longer follow up and a separate analysis considering both the overall population and patients discharged alive.

In our study, we found that one-year survival with good neurological outcome was significantly higher in the complete revascularization group when compared to the IRA-only revascularization group both when considering survival from the event of the cardiac arrest and from the patient’s discharge. Moreover, the complete revascularization strategy was found to be independently associated with better survival with good neurological outcome. Of course, this is not a randomized trial and we do not presume to draw the conclusions that only a controlled trial could provide however, it is to be noted that no randomized trials specifically designed for this aim are present in literature so far.

While it is well known that IRA revascularization leads to a better outcome [8,10], there are many concerns about myocardial revascularization of non-IRA due to procedural complications, contrast induced kidney injury and uncertainty of patients’ long-term outcome. Our study shows for the first time that complete revascularization was independently associated with a better outcome even after correcting for serum creatinine on admission, which is one of the major determinants of contrast induced nephropathy [27]. This could be related to a possible reduction of infarct size, which can favour an improvement of the left ventricular ejection fraction or a prevention of early and late myocardial infarction recurrence [28]. 

Based on these findings, it appears to be beneficial to pursue a complete revascularization strategy rather than an IRA-only strategy in patients with OHCA and multivessel coronary artery disease. Although this result appears to be in contrast with the findings of CULPRIT SHOCK trial [29], it is important to underline some key differences. The first difference is that in our study most patients achieved complete revascularization in a second procedure (78.4%) after a median time of 5 days (IQR 2.5–10.3) and only 8 patients (21.6%) underwent complete revascularization during the index procedure. On the other hand the study protocol of the CULPRIT SHOCK trial provided for immediate multi-vessel revascularization. Moreover, only half of the patients in our study were in cardiogenic shock at the time of revascularization, so our population must be considered different from that of the CULPRIT SHOCK trial, making a one-to-one comparison difficult. This is the reason why we think that it is essential to provide data exclusively from cardiac arrest patients, because they are neither classical STEMI patients nor cardiogenic shock patients. Cardiac arrest is to be considered like a separate clinical entity that is often caused by a STEMI and that could be complicated by cardiogenic shock, rather than the simple coexistence of STEMI and shock features. 

### Study Limitations

This study has some potential limitations. The first limitation is the rather small sample size, which, however, was sufficient to ensure an adequate statistical power as highlighted by our sample size calculation. However, the low number of events after hospital discharge prevented us from providing a multivariable Cox regression model. The second limitation is the single centre design of the study; therefore, our results should be confirmed by multicentre studies. The third limitation is that one may argue that complete revascularization was provided only to those patients who were expected to have better outcomes. This is not a randomised trial so the decision about the type of revascularization was left to the interventional cardiologist. However, in multivariable analysis we corrected for both possible confounders and known risk factors such as age, type of presenting rhythm and cardiac arrest duration. Moreover, through our secondary endpoint considering only those patients discharged alive, we have averted the risk of considering patients with a priori poor outcome.

## 5. Conclusions

This observational study suggests that in patients resuscitated from an out-of-hospital cardiac arrest, complete myocardial revascularization during the index hospitalization is associated with a better one-year survival with good neurological outcome compared to IRA-only revascularization. 

## Figures and Tables

**Figure 1 jcm-11-05071-f001:**
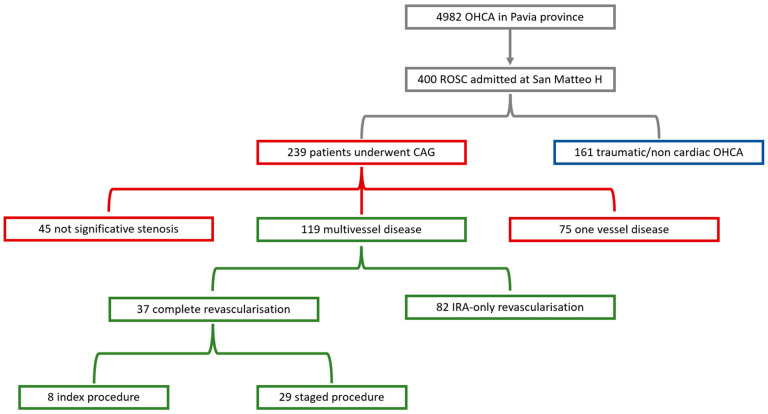
Flow-chart of the study.

**Figure 2 jcm-11-05071-f002:**
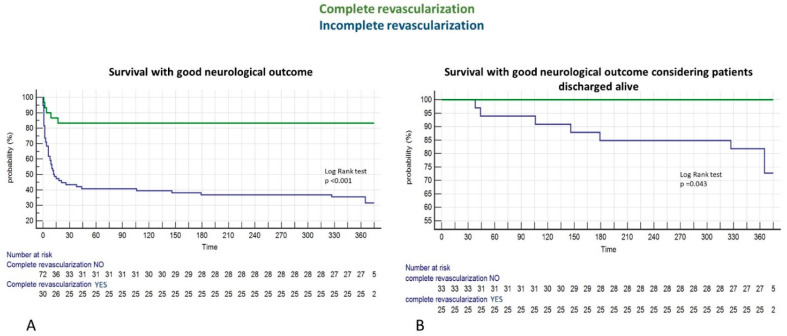
Kaplan-Meier survival curves for the overall population (**A**) and the patients discharged alive (**B**).

**Table 1 jcm-11-05071-t001:** Characteristics of patients resuscitated from OHCA who underwent a coronary angiography and with multivessel disease.

	Patients Who UnderwentCAG (*n* = 239)	Patients with Multivessel Disease (*n* = 119)
**Age (years)**	63.7 ± 12.4	66.8 ± 12
**Sex (%)**		
**Male**	191 (79.9)	98 (82.4)
**Female**	48 (20.1)	21 (17.6)
**BMI (kg/m^2^)**	25.7 (24.1–27.8)	26 (24–28)
**Hypertension (%)**	174 (72.8)	90 (81.1)
**Diabetes (%)**	38 (15.9)	19 (17.3)
**Smoker (%)**	29 (12.1)	35 (31)
**Hypercholesterolemia (%)**	123 (51.5)	67 (60.4)
**Previous myocardial revascularization (%)**		
**Previous PCI**	13 (5.4)	9 (7.5)
**Previous CABG**	31 (13.1)	18 (15.1)
**Previous PCI + CABG**	2 (0.8)	1 (0.8)
**Rhythm at presentation (%)**		
**Shockable**	201 (84.1)	100 (84)
**Not shockable**	38 (15.9)	19 (16)
**First rhythm detected (%)**		
**VF**	181 (75.7)	88 (74)
**Pulseless VT**	3 (1.3)	3 (2.5)
**PEA**	22 (9.2)	13 (10.9)
**Asystole**	12 (5.0)	5 (4.2)
**AED shockable**	17 (7.1)	9 (7.6)
**AED not shockable**	4 (1.7)	1 (0.8)
**ECG diagnostic for STEMI (%)**	162 (67.9)	75 (74.3)
**HR (bpm)**	99 ± 29	101 ± 30.3
**Median CA duration (min)**	24 (13.6–41.9)	23 (13–42)
**Time from CA to CAG (days)**	0.8 ± 3.4	0.7 ± 2.4
**LVEF (%)**	37 (30–45)	40 (30–45)
**Pharmacological haemodynamic support (%)**	80 (33.5)	37 (33)
**IABP (%)**	14 (5.9)	7 (6.4)
**ECMO (%)**	26 (10.9)	9 (8.2)
**IABP + ECMO (%)**	9 (3.7)	6 (5.5)
**Serum Creatinine on admission (mg/dL)**	1.04 (0.9–1.3)	1.03 (0.9–1.3)
**hs-TNI on admission (ng/L)**	695 (517–1126)	913 (245.8–4455.5)
**hs-TNI peak value (ng/L)**	51,194 (7853–102,181)	52,611 (9662–106,017)
**CK on admission (U/L)**	231 (132–534)	263 (133–541)
**CK peak value (U/L)**	1947 (735–5154)	2106 (985–4013)
**Hb on admission (g/dL)**	13.4 ± 1.8	13.7 ± 1.9
**Survival at discharge (%)**	157 (66)	72 (60.5)

AED, automated external defibrillator; BMI, body mass index; CABG, coronary artery by-pass graft; PCI, percutaneous coronary intervention; VF, ventricular fibrillation; VT, ventricular tachycardia; PEA, pulseless electrical activity; ECG, electrocardiogram, STEMI, ST-elevation myocardial infarction; HR, heart rate; CA, cardiac arrest; CAG, coronary angiography; LVEF, left ventricular ejection fraction; IABP, intra-aortic balloon pump; ECMO, extra-corporeal membrane oxygenation; hs-TNI, high-sensitivity cardiac troponin I; CK, creatin-kinase; Hb, hemoglobin.

**Table 2 jcm-11-05071-t002:** Characteristics of the patients comparing complete vs. IRA-only revascularization.

Characteristic	Complete Revascularization (CR) (*n* = 37)	IRA-Only Revascularization (*n* = 82)	*p* Value
**Age (years)**	62.0 ± 11.6	69.0 ± 12.1	0.16
**Sex**			
**Male**	33 (89%)	65 (79%)	0.19
**Female**	4 (11%)	17 (21%)	
**BMI (kg/m^2^)**	26.1 (24.2–29.3)	25.4 (24.2–27.8)	0.35
**Hypertension**	26 (70%)	70 (86%)	**0.04**
**Diabetes**	2 (5.4%)	19 (23%)	**0.02**
**Smoker**	13 (35%)	24 (29%)	0.63
**Hypercholesterolemia**	25 (67%)	34 (56%)	0.27
**Previous myocardial revascularization**	6 (16%)	22 (27%)	0.27
**Rhythm at presentation**			
**Shockable**	35 (95%)	65 (79%)	**0.035**
**Not shockable**	2 (5%)	17 (21%)	
**First rhythm detected**			0.23
**VF**	29 (78.4%)	59 (71.9%)	
**Pulseless VT**	1 (2.7%)	2 (2.5%)	
**PEA**	2 (5.4%)	11 (13.4%)	
**Asystole**	0 (0%)	5 (6.1%)	
**DAE shockable**	5 (13.5%)	4 (4.9%)	
**DAE not shockable**	0 (0%)	1 (1.2%)	
**ECG diagnostic for STEMI**	30 (81%)	58 (70%)	0.23
**HR (bpm)**	100 ± 34	100 ± 28	0.95
**Median CA duration (min)**	18.3 (4.0–32.7)	27.3 (14.0–42.3)	**0.05**
**Time from CA to CAG (days)**	0.5 ± 1.7	0.8 ± 2.6	0.21
**Time from CA to CR**	5 (2.5–10.3)		
**LVEF (%)**	40 (37–45)	40 (25–50)	0.46
**Pharmacological haemodynamic support**	8 (22%)	32 (39%)	0.09
**LV mechanical assistance**	6 (16.2%)	17 (20.1%)	0.09
**IABP**	2 (5%)	5 (6.1%)	
**ECMO**	0	10 (12%)	
**IABP + ECMO**	4 (10%)	2 (2.4%)	
**Number of vessels involved**			0.45
**Two-vessel disease**	22 (59.5%)	47 (56.1%)	
**Three-vessel disease**	15 (40.5%)	36 (43.9%)	
**Chronic total occlusion**	9 (24.3%)	41 (50.0%)	**0.009**
**Serum Creatinine on admission (mg/dL)**	0.98 (0.83–1.05)	1.1 (0.9–1.4)	**0.019**
**TNI on admission (ng/L)**	581 (238–1316)	1866 (260–6972)	0.13
**TNI peak value (ng/L)**	56,925 (10,258–188,560)	51,180 (8377–89,998)	0.45
**CK on admission (U/L)**	177 (118–349)	318 (151–633)	**0.03**
**CK peak value (U/L)**	2310 (1201–5388)	1869 (765–3301)	0.49
**Hb on admission (g/dL)**	14.4 ± 1.5	13.1 ± 1.9	**0.0005**
**Survival at discharge**	32 (86.5%)	41 (50.6%)	**0.0002**

BMI, body mass index; CABG, coronary artery by-pass graft; PCI, percutaneous coronary intervention; VF, ventricular fibrillation; VT , ventricular tachycardia; PEA, pulseless electrical activity; ECG, electrocardiogram, STEMI, ST-elevation myocardial infarction; HR, heart rate; CA, cardiac arrest; CAG, coronary angiography; CR, complete revascularization; LV, left ventricle; LVEF, left ventricle ejection fraction; IABP, intra-aortic balloon pump; ECMO, extra-corporeal membrane oxygenation; TNI, troponin I; CK, creatin-kinase; Hb, hemoglobin.

**Table 3 jcm-11-05071-t003:** Univariable and multivariable logistic regression analysis logistic regression analysis for the probability of receiving a complete revascularization.

Variable	Univariable Logistic Regression Analysis for the Probability of Receiving a Complete Revascularization			Multivariable Logistic Regression Analysis for the Probability of Receiving a Complete Revascularization		
	Odds Ratio	CI 95%	*p* Value	Odds Ratio	CI 95%	*p* Value
**Age**	0.98	0.94–1.01	0.20			
**Diabetes**	0.3	0.08–1.1	0.07			
**Previous myocardial infarction**	0.44	0.16–1.2	0.11			
**Previous myocardial revascularization**						
**CABG**	1.18	0.3–5.3	0.83			
**PCI**	0.39	0.1–1.5	0.16			
**Shockable rhythm at presentation**	5.1	1.12–22.9	**0.03**	2.7	0.5–13.5	0.23
**Cardiac arrest duration (min)**	0.4	0.2–0.9	**0.02**	0.99	0.97–1	0.34
**Two versus three-vessels disease**	1	0.5–2.2	0.98			
**Chronic total occlusion**	0.37	0.16–0.85	**0.02**	0.4	0.2–1	**0.049**
**SCr on admission (mg/dL)**	0.16	0.04–0.7	**0.018**	0.3	0.1–1.6	0.17
**TNI on admission (ng/L)**	0.7	0.5–1.1	0.17			
**TNI peak value (ng/L)**	1.4	0.8–2.5	0.3			
**CK on admission (U/L)**	0.4	0.1–1	0.06			
**CK peak value (U/L)**	1.4	0.5–4.3	0.52			
**Pharmacological or mechanical circulatory support**	0.9	0.3–2.6	0.86			

CABG, coronary artery by-pass graft; PCI, percutaneous coronary intervention; SCr, serum creatinine; TNI, troponin I; CK, creatin-kinase.

**Table 4 jcm-11-05071-t004:** Univariable and multivariable Cox Regression model for the probability of death or poor neurologic outcome.

	Univariable Cox Regression Model for the Probability of Death or Poor Neurologic Outcome			Multivariable Cox Regression Model for the Probability of Death or Poor Neurologic Outcome		
Variable	HR	95%CI	*p*	HR	95%CI	*p*
**Age (year)**	1	0.99–1.02	0.34			
**Cardiac arrest duration (min)**	1.02	1.01–1.02	<0.001	1.02	1.01–1.04	0.04
**Shockable presenting rhythm**	0.3	0.2–0.4	<0.001	0.35	0.1–0.9	0.03
**Complete revascularization**	0.2	0.1–0.4	<0.001	0.33	0.1–0.9	0.04
**Number of vessels involved**						
**0**	Ref					
**1**	0.6	0.4–1.1	0.13			
**2**	0.6	0.3–1.1	0.12			
**3**	1.3	0.7–2.2	0.42			
**LVEF > 40%**	0.8	0.4–1.4	0.37			
**CTO**	1.3	0.8–1.9	0.30			
**Previous revascularization**						
**No**	Ref					
**CABG**	0.8	0.3–1.9	0.6			
**PCI**	1.5	0.8–2.7	0.18			
**Previous STEMI**	1.2	0.7–1.9	0.5			
**TNI (peak) (ng/L)**	1.6	1–2.4	0.04	1.3	0.7–2.6	0.41
**CK (peak) (U/L)**	1.8	1–3.2	0.05			
**SCr on admission (mg/dL)**	1.3	1–1.7	0.016	0.9	0.5–1.7	0.76
**Pharmacological or mechanical circulatory support**	2.8	1.8–4.4	<0.001	1.6	0.6–4.1	0.33

CABG, coronary artery by-pass graft; PCI, percutaneous coronary intervention; STEMI, ST-elevation myocardial infarction; LVEF, left ventricular ejection fraction; TNI, troponin I; CK, creatin-kinase; SCr serum creatinine.

## Data Availability

Data are available upon request to the corresponding author.

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
