# Peer review of "Complete Revascularization and One-Year Survival with Good Neurological Outcome in Patients Resuscitated from an Out-of-Hospital Cardiac Arrest"

_jcm, 2022, doi:10.3390/jcm11175071_

Round 1

Reviewer 1 Report

First of all, I would like to thank you for the opportunity to review an interesting paper by Kajana et al on the evaluation of Complete revascularization and one-year survival with good neurological outcome in patients resuscitated from an out-of-hospital cardiac arrest. In this work, the authors showed that in patients resuscitated from an out-of-hospital cardiac arrest, complete myocardial revascularization during the index hospitalization is associated with a better one-year survival with good neurological outcome compared to IRA-only revascularization. Without a doubt, these results are interesting and important for practitioners, but when reviewing the manuscript, I had a number of comments.

1.     Primarily, attention is drawn to the fact that in the group of incomplete revascularization survival at discharge was significantly lower (50.6%) than in the group of complete revascularization (86.5%). Differences in this indicator between groups were the highest among all factors studied by the authors (p=0.0002). Several questions inevitably arise from this fact. If the matter is primarily in the absence of complete revascularization (as the authors of the article argue), then how ethical was it to continue this study with an assessment of annual results, and not to stop it when such results of inpatient treatment were obtained. It is good that the study was registered with ClinicalTrial.gov, but was there an analysis of the interim data? Secondly, why the interventional cardiologist alone made the decision on the tactics of revascularization without the participation of the cardio team? Thirdly, as far as I understood from the text of the article, revascularization was limited only to PCI, while at the same time, in chronic occlusions, the frequency of complete revascularization was lower. Does the clinic have a department of cardiovascular surgery, does it perform CABG for acute coronary syndrome, and does a cardiac surgeon take part in deciding on the tactics of revascularization? It is possible that CABG was indicated for some patients with complete occlusion of the coronary artery.

2.     In my opinion, in the group of survivors after inpatient treatment, a multiple logistic regression analysis of factors associated with one-year survival with good neurological outcome should also be performed. The results of Kaplan-Meier survival curves for the patients discharged alive (Figure 2B) presented by the authors do not convince that it was complete myocardial revascularization that had an independent effect on these results, since the groups also differed in other parameters.

3.     Finally, I understand that it is sometimes difficult to keep up with current publications in the vast medical literature, but still. More recently, in the article by Kim et al. (see link) examined a much-related topic - the impact of total myocardial revascularization in survivors of out-of-hospital cardiac arrest on hospital outcomes. Yes, this study is limited to a 30-day follow-up period, but it has not a number of other limitations of the peer-reviewed article (the study is multicenter; patients underwent revascularization not only with PCI, but also with CABG). In any case, this article should be discussed in the peer-reviewed article both in the introduction and in the discussion.

References:

Kim YJ, Park DW, Kim YH, Choi M, Kim SJ, Lee GT, Lee DH, Lee BK, Oh JS, Oh SH, Lee DH, Kim WY. Immediate complete revascularization showed better outcome in out-of-hospital cardiac arrest survivors with left main or triple-vessel coronary diseases. Sci Rep. 2022 Mar 14;12(1):4354. doi: 10.1038/s41598-022-08383-x.

Author Response

We attached the reply to reviewer as a word document

Reviewer 2 Report

I would like to thank the Editor for the opportunity to review this paper by Italian colleagues; the paper is interesting, gathers equally interesting data, and focuses on a topic that is of extreme interest. Most of the corrections I point out are of minor importance, but I have some major concerns that necessarily needs to be addressed.

MAJOR CONCERNS:

First:

Authors describe (lines 83-85) the following: "the decision to perform a complete revascularisation or an IRA-only revascularisation during the index procedure was left to the interventional cardiologist".  Table 3 shows that there is an inverse correlation between cardiac arrest and complete revascularisation (OR 0.4, p 0.02), as well as for chronic lesions (OR 0.37, p 0.02), while there is a direct correlation between having a defibrillable rhythm at cardiac arrest presentation and complete revascularisation (OR 5.1, p < 0.03).

Now, since the choice of interventional cardiologist is not randomised, but is based on clinical judgement, it is possible to hypothesize that when the patient has had a cardiac arrest, the interventional cardiologist is less likely to perform a complete revascularisation. Therefore, the choice is not randomised, but based on clinical judgement. The logical step linking the outcome to the clinical gesture (CR vs. IRA) is vitiated by the fact that the choice of the clinical gesture is not randomised, but probably linked to the patient's clinical condition. This element of discussion must be integrated and put: 1. ABSOLUTELY in the discussion, 2. in the limits (it is already written), 3. in the general conclusions (and in those of the abstract). I know that the Authors have written down this element in the limits, but this limit is so important that it cannot lead to a conclusion such as the one described.

Second:

In table 3, the references to units of measurement (min, mg/dl, etc...) should be removed; moreover, it should be explained what an inverse correlation (OR 0.16) between creatinine and complete revascularisation means. The same applies to the other serum values: the unit of measurement must be removed (Authors talk about an odd ratio) and it must also be explained what it means.

Third:

Endpoints are to be inserted after definitions; also, a very important point: are there patients alive and not discharged? Please clarify this difference between outcomes of the living at one year and outcomes of the living and discharged at one year. When we talk about outcomes, we dwell a lot on descriptive statistics, while primary and secondary outcomes come much later. Again, are Authors sure that the primary and secondary outcomes described in the M&Ms are maintained in the Results section? To evade this concern, Authors need to: a) better clarify primary/secondary outcomes, b) keep them the same in the Results, c) describe them as early as possible in the Results.

Fourth:

What about Informed Consent? This must be extensively described in the M&M section.

MINOR CONCERNS:

All abbreviations, the first time they are reported, are to be described. In the abstract there is no explanation of IRA, CAG; CR on line 184 has no previous explanation in the text, but they are in the table, etc...

There are sentences that need to be better written in English, such as "it has already been demonstrated" (line 44, in my opinion an element to remove) or "on the contrary" (line 187, to be replaced with "on the other hand"), etc... These are phrases that betray a Latin language derivation, but are not phrases commonly used in English. An extensive review of English is strongly recommended. The purpose of the study (68-71) should be described a little better as English.

In M&M (lines 80-81), replace the name of the hospital with the type of hospital (primary care, tertiary hospital, etc.) which is more informative for the reader. The name can be kept, but in brackets.

The paragraph "sample size" should be inserted in the statistical part, before the descriptive statistics.

Lines 169-170 should be enhanced. Table 1 is interesting if (and only in this case), they are representative of patients with multivessel disease. In my opinion, Authors are two possibilities: either lines 169-170 explicitly state that the patients with multivessels disease are distributed equally to the total patients, or - even better - the patients who are involved in the therapeutic choice CR vs. IRA are put in table 1. The reader is interested in multivessels disease patients, not to all patients.

Figure 2: In the KMs Authors have to put the progressive number of patients in the x axis; furthermore, Authors have to clarify what is meant by "survival probability": is it a probability or certain data? As mentioned earlier, also, what is the difference between A and B? It must be assumed that patients who remained in hospital were also considered.

Author Response

(The authors gave the same response as above.)

Round 2

Reviewer 1 Report

The authors have done a good job of correcting the manuscript, I have no fundamental comments left. However, sources 23 and 24 in the list of references need to be supplemented with bibliographic data.

Author Response

We thank the reviewer for highlighting this issue regarding references 23 and 24. We have corrected it as suggested.